# Influence of Functional Traits of Dominant Species of Different Life Forms and Plant Communities on Ecological Stoichiometric Traits in Karst Landscapes

**DOI:** 10.3390/plants13172407

**Published:** 2024-08-28

**Authors:** Yang Wang, Limin Zhang, Ling Feng, Zuhong Fan, Ying Deng, Tu Feng

**Affiliations:** 1School of Ecological Engineering, Guizhou University of Engineering Science, Bijie 551700, China; fanzh961226@163.com (Z.F.); dy3594142@163.com (Y.D.); fengtu@gues.edu.cn (T.F.); 2Guizhou Province Key Laboratory of Ecological Protection and Restoration of Typical Plateau Wetlands, Bijie 551700, China; 3Institute of Mountain Resources of Guizhou Academy of Sciences, Guiyang 550025, China; zhanglimin563406@163.com; 4College of Life Sciences, Guizhou University, Guiyang 550025, China; 15870149772@163.com

**Keywords:** life forms, plant community, functional traits, ecological stoichiometry, karst

## Abstract

Assessing the functional traits and ecological stoichiometric characteristics of dominant species across different life forms within plant communities in karst environments and investigating the inherent connection between them can provide insights into how species adjust their functional attributes in response to habitat heterogeneity. This approach offers a more comprehensive understanding of ecosystem processes and functions in contrast to examination of the taxonomic diversity of species. This study examines the relationship between the functional characteristics of dominant species in plant communities of various life forms in karst environments, focusing on deciduous leaf–soil ecological stoichiometry. The investigation relies on community science surveys, as well as the determination and calculation of plant functional traits and ecological stoichiometries, in plant communities of various life forms in Guizhou (a province of China). The findings of our study revealed considerable variability in the functional trait characteristics of dominant species across different plant-community life forms. Specifically, strong positive correlations were observed among plant height (PLH), leaf area (LA), leaf dry matter content (LDMC), and specific leaf area (SLA) in the dominant species. Additionally, our results indicated no significant differences in leaf ecological stoichiometry among different life forms. However, we did observe significant differences and strong positive correlations between soil N:P, withered material C:N, and apomictic C:P. Furthermore, our study found that plant height (PLH), leaf area (LA), and specific leaf area (SLA) were particularly sensitive to the ecological stoichiometry of soil and apomixis. The results of our study suggest that the functional traits of diverse plant-community life forms in karst regions are capable of adapting to environmental changes through various expressions and survival strategies. The development of various plant-community life forms in karst areas is particularly vulnerable to phosphorus limitation, and the potential for litter decomposition and soil nutrient mineralization is comparatively weaker. The functional traits of various plant-community life forms in karst regions exhibit greater sensitivity to both the soil’s C:N ratio and the C:N ratio of apomictic material. Habitat variations may influence the ecological stoichiometric characteristics of the plant leaf–apomictic soil continuum.

## 1. Introduction

As ecology continues to evolve, plant functional traits have proven to be an important means of exploring various ecological frontiers [1]. Plant functional traits serve as a crucial link between plants and their environment [2] and also reflect the variability of species in terms of growth, survival, and reproduction. This statement contributes to a more comprehensive understanding of the distribution of plant species, the process of community formation, and ecosystem function [3]. Presently, there is a substantial body of research focused on plant functional traits, which includes the examination of leaf, branch, and stem traits, as well as root traits, in order to elucidate the relationships among these traits [4]. Moreover, it is essential to investigate the correlation between plant characteristics and the environment in order to comprehend the mechanisms of coexistence at different spatial and temporal levels and to analyze the dynamics of species in reaction to environmental fluctuations. Secondly, it is essential to investigate the correlation between plant characteristics and the environment in order to comprehend the coexistence mechanisms at different spatial and temporal levels and to analyze the dynamic attributes of species in reaction to environmental fluctuations [5]. The functional traits of plant leaves have been shown to be closely associated with plants’ capacity to utilize resources and, to some extent, can indicate their survival strategies for adapting to environmental changes [6]. The examination of functional traits in plant leaves can enhance our comprehension of how leaf physiology reacts to environmental fluctuations [7]. Various indicators of plant functional traits, such as specific leaf area, leaf nitrogen content, leaf thickness, leaf dry matter content, chlorophyll content, leaf relative water content, stem tissue density, leaf carbon content, stem nitrogen content, stem phosphorus content, and other traits, are commonly selected [8,9,10].

Ecological stoichiometry pertains to the proportion of vital chemical elements implicated in ecological interactions and processes [11]. The interconnected organic systems of plants and soils necessitate the analysis of their carbon (C), nitrogen (N), and phosphorus (P) contents and ratios, which are essential for the energy cycle and stability of ecosystems [12]. The southwestern karst region is considered to be one of the most ecologically vulnerable areas in China, and the ecological and environmental challenges have emerged as a bottleneck constraining the economic and social development of the region [13]. Ecological chemometrics combines the basic principles of biology, chemistry, and physics, including those of ecology and chemometrics, taking into account the first law of thermodynamics, the principle of natural selection for biological evolution, and the theory of the central law of molecular biology [14]. The rapid advancement of ecological chemometrics has attracted growing attention from scholars, who are exploring various aspects of this technique. Previous studies have documented the ecological and chemometric properties of soils in diverse karst ecosystems [15,16]. Recently, a study was conducted to analyze the eco-chemical quantification of leaves from dominant species in secondary karst forests. The C:N:P stoichiometry of the leaf–deciduous leaf–soil continuum in secondary forests was calculated [17]. Carbon (C), nitrogen (N), and phosphorus (P) are essential elements for life, with nitrogen and phosphorus serving as the primary limiting factors in natural terrestrial ecosystems [18]. The contents of these elements have the potential to influence the species composition and productivity of plant communities. The biogeochemical processes of systems are responsible for regulating the cycling of nutrients, as indicated by reference [19]. Hence, carbon (C), nitrogen (N), and phosphorus (P) are crucial in ecosystem processes and serve as connections between community species diversity, plant functional traits, and ecological stoichiometry [20]. Utilizing this framework to investigate the inherent connection of ecosystem processes and their feedback effects can lead to a deeper understanding of ecosystem nutrient cycling principles and system stabilization mechanisms. Variations in leaf traits among different life forms lead to differences in the requirement for and utilization of light, precipitation, temperature, nutrients, and other resources [21]. These variations also result in a wide range of adaptations to different topographic conditions [22]. Hence, it is essential to examine the relationship between functional traits and ecological stoichiometry in various plant life forms in order to comprehend the mechanisms underlying biodiversity maintenance and the survival tactics of plants.

Plant functional traits play a crucial role in illustrating the connection between plants and their environment, revealing the adaptive strategies and regulatory mechanisms that plants employ in particular environments [23]. The capacity of plants to obtain and utilize resources is also pertinent. The stoichiometric characteristics of elements such as carbon (C), nitrogen (N), and phosphorus (P) in organisms are closely linked to their stability, community species composition, and biogeochemical cycling processes. These characteristics serve as potent instruments for examining nutrient restriction and cycling [24]. The functional traits and ecological stoichiometric characteristics of forest plants are subject to influence by environmental factors, which in turn constrain ecosystem processes and functions. Consequently, an examination of their fundamental characteristics and inherent relationships can lay the groundwork for the sustainable management of forest ecosystems.

Based on this premise, this paper aims to uncover the functional traits and ecological stoichiometric characteristics of various life forms within plant communities in karst regions and to finally clarify the interaction between soil and vegetation. This will be achieved through community surveys and the measurement or calculation of plant functional traits and ecological stoichiometric indexes. The ultimate goal is to offer insights for research on the mechanisms involved in the establishment of forest communities and the preservation of biodiversity in karst environments.

## 2. Research Methods

### 2.1. Study Area

The Huaxi District is located in the southern region of Guiyang City, Guizhou Province, in China (106°27′–106°52′ E, 26°11′–26°34′ N), within the watershed area that separates the Yangtze River system and the Zhujiang River system. The total land area encompasses 953.83 km^2^, with 94% of the territory exhibiting a karstic landscape. This includes a mountainous and hilly topography, delicate habitats, and a low land-carrying capacity. Due to its elevated plateau, the region experiences a highland monsoon humid climate, characterized by an average annual rainfall of 1178.3 mm and an average annual temperature of 14.9 °C (Figure 1).

#### 2.1.1. Soil Environment

The study area is situated in the hilly regions of Guizhou and Zhongshan on the Yunnan–Guizhou Plateau. The terrain is characterized by higher elevations in the southwest and lower elevations in the northeast. Denuded hills are interspersed with basins, valleys, and depressions, with a relative height difference of 100–200 m. The geological structure is primarily marked by north–south-trending structures, with rock layers in Huaxi District occurring in a monoclinal form. Large folds or faults are absent from the surface of the study area, indicating a relatively simple structural complexity. The main exposed strata in Huaxi District consist of the Quaternary and Middle Triassic Yangliujing Formation, with uniform rock strata and no structural passage. However, due to regional structural influences, the rock mass is relatively fractured. The predominant soil type in the study area is yellow soil, which is the most prevalent zonal soil type. Yellow soil is formed under warm and humid subtropical monsoon bioclimate conditions, with predominantly acidic pH levels, followed by neutral and less alkaline characteristics [25].

#### 2.1.2. Description of Karst Vegetation

Three distinct plant communities in the region, namely, the grass–shrub plant community (CG), the shrub plant community (GM), and the tree–shrub plant community (QG), were selected for the study based on the findings of previous research [26]. Two equations were utilized to ascertain the predominant species of trees, shrubs, and herbs. The final dominant species were identified as follows: dominant tree species include *Lindera communis*, *Itea yunnanensis* Franch., *Quercus fabri*, and chestnut *Castanea mollissima* Blume, among others, while the dominant shrub species mainly consist of *Rhamnus leptophylla*, *Rosa multiflora*, *Glochidion puberum*, *Pyracantha fortuneana*, *Coriaria nepalensis*, *Sarcococca ruscifolia*, *Rosa cymosa*, *Rhamnus heterophylla*, *Viburnum foetidum* var *ceanothoides*, and *Myrsine africana*. The dominant herbaceous plant species mainly include *Imperata cylindrica*, *Senecio asperifolius*, *Erigeron canadensis*, *Carex capilliformis*, and *Ficus tikoua*.

### 2.2. Sample Setup

Three sample plots were chosen as replicates for each plant-community life type in this study, resulting in a total of nine standard sample plots.

The research team conducted a survey and sampled plant communities between April and August 2021. In accordance with the principles of representativeness, typicality, and consistency, the research sample plots were chosen based on areas with similar slope positions, slope directions, and other land conditions. The sample plots were established at a size of 10 m × 10 m for the herbaceous stage and 30 m × 30 m for the other stages. Small sampling squares were designated within the sample plots to aid in conducting plant surveys.

Nine small sample plots, each measuring 10 m × 10 m, were established within the arborvitae community. A single 4 m × 4 m shrub-layer sample plot was established within each tree plot. A 1 m × 1 m herb-layer sub-sample square was positioned within each shrub-layer sub-sample square. Consequently, there were nine sample plots designated for the tree layer, nine for the shrub layer, and nine for the herb layer within the tree–shrub plant community. Given the presence of three replicate sample plots, the total count of sample plots within the arboreal–shrub community amounted to 81.

Nine small sample squares, each measuring 4 m × 4 m in area, were established in both the grass–shrub plant community and the shrub plant community to represent the shrub layer. A 1 m × 1 m herbaceous-layer sample plot was established within each shrub-layer sample plot. Consequently, there were nine sub-samples of shrubs and nine sub-samples of herbaceous plants in both the grass–shrub plant community and the shrub plant community. Therefore, there were a total of 36 small sample squares within these two plant communities. The two plant communities comprised a total of 108 small sample squares, with each community having 3 replicate sample plots.

In total, three plant communities with varying life forms were investigated, encompassing 81 herbaceous sample plots, 81 shrub sample plots, and 9 tree sample plots. The study ultimately documented the nomenclature, height, diameter at breast height, and crown width of both tree and shrub species. The study also recorded the names, quantities, average heights, and ground coverage of herbaceous plants to assist in the assessment of the functional traits of dominant species.

### 2.3. Sample Collection and Processing

The measurement of the plant sample was divided into two parts. The initial step entailed the selection of all plants within the sampling area, followed by the use of high pruning shears to cut the branches of the sampled plants in the four directions of the crown (south, east, north, west). Approximately 20 healthy and disease-free leaves were then collected from each branch to form a mixed sample. The second step involved selecting the top three significant values (dominant species) within the sampling area as the sampled plants and combining each dominant species from each stage into a single sample. The importance value of the tree stratum was determined by aggregating the relative multiplicity, the relative frequency, and the relative significance, then dividing the sum by 3. Similarly, the significance of the shrub and herb stratum was calculated by combining the relative abundance, the relative frequency, and the relative cover, then dividing by 3. 

A total of 27 collection nets, each measuring 1 m × 1 m, were deployed across three plant communities, with 9 nets in each community. The samples were collected in September 2021 and retrieved in March 2022. The “S” sampling method [27] was utilized to collect soil samples from the 0–20 cm depth range in each plant community. The samples were obtained from a limited sampling area (less than 20 cm, based on the actual depth) and were subsequently combined in equal volumes to form a composite sample. Several leaf samples were collected from the initial section, and 27 leaf samples, 27 deadwood samples, and 27 soil samples were collected from the subsequent section. The plant and litter samples were subjected to initial heating at 105 °C for 2 h, followed by drying at 75 °C until a constant weight was achieved. The soil samples were dried at room temperature in an indoor environment. All samples were finely ground, sieved through a 60-mesh sieve, and stored in a well-ventilated area for nutrient analysis.

### 2.4. Determination of Functional Properties and Analysis of Samples

Six metrics were selected to measure plant functional traits, comprising plant height (PLH), leaf thickness (LT), chlorophyll content (CHL), leaf dry matter content (LDMC), leaf area (LA), and specific leaf area (SLA). The approach for determination was established according to the recently developed manual for standardized measurement of global plant functional traits [28]. The fresh weight of the leaves from each plant sample was determined. The samples underwent baking at 60 °C for 72 h until they achieved a consistent weight. The measurement of the leaves’ dry weight was conducted. LDMC is determined by dividing the dry weight of leaves by the fresh weight of leaves. Leaf thickness was assessed with electronic vernier calipers (Deli, DL91150, Qingdao, China), whereas leaf length and area were determined through scanning and subsequent calculations using an HP scanner in conjunction with Photoshop software 2023 (HPScanJetN92120, Wuhan, China). The specific leaf area (SLA) was determined by dividing the leaf area by the leaf dry weight. The chlorophyll content of the leaves was assessed using a chlorophyll meter (Linde, LD-YD, Jining, China). Plant leaf area (LA) was determined through direct field measurements, while leaf dry matter content (LDMC) was calculated as the ratio of the leaf’s weight after drying to its fresh weight. The organic carbon content (Soil_C) was quantified using the potassium dichromate oxidation–external heating method [29]. Plant and litter samples underwent digestion using the H_2_SO_4_-H_2_O_2_ method. The total nitrogen content in litter and leaves was assessed through the indophenol blue colorimetric method (Litter_C, Leaf_C) (NY/T2017-2011) [30], whereas the total phosphorus content in litter was determined using the molybdenum antimony colorimetric method (Litter_P, Leaf_P) (NY/T2017-2011) [30]. Soil samples underwent testing for total nitrogen utilizing the Kjeldahl nitrogen determination method (Soil_N), as per the LY/T1228-2015 standard [31], and for total phosphorus using the NaOH melting–molybdenum antimony colorimetric method (Soil_P), in accordance with LY/T1232-2015 [32].

### 2.5. Data Processing

The data were initially organized using Microsoft Excel 2019. Prior to analysis, the data underwent normality testing and were then subjected to ANOVA. The data were analyzed utilizing SPSS 25.0 statistical software [33]. One-way analysis of variance (ANOVA) and Tukey’s honestly significant difference multiple-comparisons test were utilized to evaluate the differences in functional trait values, leaf-litter–soil nutrient content, and ecological stoichiometry across different life forms within plant communities. Pearson correlation analysis was employed to elucidate the association between these metrics, and the data were reported as means ± standard deviations. To further examine the patterns of variation in functional traits and leaf litter–soil stoichiometry, multivariate analysis was conducted using the “ggcor”, “vegan”, “dplyr”, and “ggplot2” packages in R version 4.3.2. Software packages in the R 4.3.2 programming language were used for psychological analysis [34].

## 3. Results and Analysis

### 3.1. Characterizing Changes in Functional Traits of Dominant Species in Plant Communities with Different Life Forms

As illustrated in Figure 2, there was notable variation in the functional traits of dominant species across different plant-community life forms. The plant leaf area index (PLH) exhibited significant variation among different life forms within karst plant communities, with tree–shrub communities demonstrating the highest values, followed by shrub communities and grass–shrub communities. Meanwhile, there was no significant difference in leaf thickness (LT) between grass–shrub and shrub communities. However, leaf thickness was significantly greater in shrub communities compared to tree–shrub communities. Moreover, the leaf area (LA) exhibited a statistically significant increase in tree–shrub communities compared to grass–shrub communities. No significant differences were observed in LDMC, SLA, or CHL among plant communities with different life forms. The correlation analysis of plant functional traits indicated a strong positive correlation between leaf area (LA) and plant height (PLH), as well as between leaf area (LA) and leaf dry matter content (LDMC). The study found a significant positive correlation between PLH and LDMC, as well as a significant positive correlation between specific leaf area (SLA) and LA. Furthermore, there was a notable inverse relationship between SLA and chlorophyll (CHL) and LDMC (Figure 3). 

### 3.2. C, N, and P Contents of Leaf Litter and Soil of Dominant Species in Plant Communities with Different Life Forms

The carbon (C), nitrogen (N), and phosphorus (P) contents of leaf litter and soil for dominant species in plant communities with different life forms indicate that there were no significant differences in Leaf_C or Leaf_P among the various plant communities. However, Leaf_N exhibited notable variations across the various life forms, with arborvitae demonstrating the highest values, followed by shrubs and then grass–shrubs (Figure 4). Moreover, the Litter_C content was notably higher in the shrub areas in comparison to the grassy shrub areas. Significant variations in Litter_N and Litter_P were observed among different life forms within arboreal communities, with the ranking being grass–shrub > shrub > arboreal communities. Significant differences were observed in Soil_C and Soil_N among various life forms of plant communities, with the order being shrub > grass–shrub > arboreal communities. Furthermore, Soil_P displayed notable variations among various life forms within plant communities, with the ranking being grass–shrub community > shrub community > tree–shrub community.

### 3.3. Characterization of Changes in Leaf-Litter–Soil Ecological Stoichiometry of Dominant Species in Plant Communities with Different Life Forms

The investigation into the variations in the leaf-litter–soil ecological stoichiometry of dominant species in plant communities with diverse life forms indicated a significant difference in Leaf_C.N between grass–shrub (58.35) and tree–shrub (27.22) communities. However, the leaf ecological stoichiometry showed no significant differences among dominant species in plant communities with different life forms. Additionally, Leaf_N.P (CG—16.27, GM—17.33, QG—17.81) and Leaf_C.P (CG—625.68, GM—602.56, QG—643.59) did not display significant variations across plant communities with different life forms. Conversely, Litter_C.N (CG—16.68, GM—24.25, QG—19.92) and Litter_C.P (CG—362.34, GM—478.86, QG—436.22) consistently exhibited notable variances across various plant-community types. Specifically, the shrub community displayed significantly elevated levels compared to the tree–shrub community, while the tree–shrub community exhibited significantly higher levels than the grass–shrub community. Litter biomass, however, showed no significant variations across various types of plant communities. Moreover, Soil_N.P demonstrated notable variations across various plant-community types, with the shrub community (6.92) showing superior performance compared to the tree–shrub (6.02) community and the latter outperforming the grass–shrub (4.83) community. The value of N exhibited significant variation across various plant communities, with the highest levels found in shrub communities, followed by arboreal–shrub communities and then grass–shrub communities. Furthermore, Soil_C.P was notably elevated in the shrub (84.75) and tree–shrub (88.64) communities in comparison to the grass–shrub (34.29) community (Figure 5). Further examination of the ecological stoichiometry of leaf litter and soil for dominant species across various plant-community life forms uncovered notable correlations. Specifically, Leaf_N.P demonstrated a highly significant positive correlation with Leaf_C.P, whereas Litter_C.N displayed a highly significant positive correlation with Litter_C.P. Furthermore, there were highly significant positive correlations observed between Soil_C.N, Soil_C.P, and Litter_C.P, and Soil_N.P showed a highly significant positive correlation with Soil_C.P (Figure 6).

### 3.4. Relationships between Functional Traits of Dominant Species and Deciduous Leaf-Litter–Soil Ecological Stoichiometry in Plant Communities with Different Life Forms

The relationships between the functional traits of dominant species in various life forms within plant communities and the ecological stoichiometry between deciduous leaf litter and soil were examined. The results revealed that specific leaf area (SLA) was significantly positively correlated with Soil_N.P; plant height (PLH) exhibited a significant positive correlation with Litter_C.N, Litter_C.P, Soil_C.P, and Soil_C.N; leaf area (LA) showed a significant positive correlation with Litter_C.N; and Soil_C.N displayed a significant positive correlation. Additionally, leaf dry matter content (LDMC) was found to have a significant positive correlation with Soil_C.N (Figure 7).

To examine the influence of dominant species’ functional traits on the ecological stoichiometry of plant communities across various life forms, we categorized the plant functional shapes into two groups, namely, “Fun01” and “Fun02”. In this study, the three functional traits, PLH, LT, and LA, were collectively referred to as “Fun01” traits, signifying notable variations in plant communities across different life forms. The three functional traits, LDMC, SLA, and CHL, were also taken into consideration as “Fun02” traits, indicating no significant variations in plant communities across different life forms. The findings indicated that the characteristics of class A had a notable effect on the ratio of carbon to nitrogen in the soil and litter, whereas the traits of class B exerted a substantial influence on the ratio of nitrogen to phosphorus in the soil and the ratio of carbon to phosphorus (Figure 8). Mantel analysis revealed that soil ecological stoichiometry significantly influenced the functional traits of plant communities with diverse life forms in karst environments. Furthermore, the ecological stoichiometry of leaves and litter had a significant impact on the functional traits of plant communities with varying life forms in karst environments. The plant communities in karst areas exhibited diverse responses to functional traits across different life forms, while the ecological stoichiometry of leaves and litter remained unaffected by these responses. Through Mantel analysis, we conducted a comprehensive investigation to explore the relationship between plant functional traits and ecological stoichiometry. The results of our study indicated notable positive associations between plant leaf height (PLH) and soil C:N, as well as litter C:N. Moreover, leaf thickness (LT) displayed significant negative correlations with soil C:N, while leaf area (LA) exhibited significant positive correlations with soil C:N and litter C:N. Additionally, specific leaf area (SLA) showed significant positive correlations with soil N:P and soil C:P (Figure 9). The diagram depicts relationships. The ratio of carbon to nitrogen in the soil is expected to be a significant factor affecting plant leaf height, leaf thickness, and leaf area.

## 4. Discussion

### 4.1. Changes in Functional Traits of Dominant Species in Plant Communities with Different Life Forms in Karst Areas

Its life form is a characteristic reflection of a plant’s physiological, structural, and external morphology with a certain degree of stability after long-term adaptation to an integrated habitat [35]. In this study, we found that LDMC, SLA, and CHL showed no significant differences among different life forms. PLH exhibited significant differences among different life forms. LT showed significant differences between the shrub community and the arboreal–shrub community, as well as between the grass–shrub community and the arboreal–shrub community. LA displayed significant differences between the grass–shrub community and the arboreal–shrub community. These findings align with Yao et al.’s research and indicate that in the same environment, plants with different life forms adopt distinct leaf traits to adapt to their surroundings [36]. In addition, PLH was significantly higher in the arborvitae community than in the shrub and grass–shrub communities (Figure 2A). LT was significantly lower than in the shrub and grass–shrub communities, and LA was significantly higher than in the other life forms (Figure 2B,D). These results are consistent with the findings of Kong et al., suggesting that the competitiveness, productivity, and restoration of vegetation in the arborvitae and grass–shrub communities were stronger compared to those in the shrub and grass–shrub communities. The grass–shrub communities were found to be in more severe environments in terms of life forms, and leaves could better adapt to the environment by increasing their thickness. In the grass–shrub community, the leaf blades could better adapt to the environment and retain soil and water by increasing their thickness, thus enabling them to resist the harsher environment [37]. In addition, the tree–shrub community could improve light and water use efficiency by reducing leaf thickness and increasing leaf area. This adjustment also helps enhance transpiration and promote the rapid growth of plants [38].

### 4.2. Changes in C, N, and P in the Leaf-Litter–Soil Ecological Stoichiometry of Dominant Species in Plant Communities with Different Life Forms in Karst Areas 

Leaf C:N and C:P ratios can characterize a plant’s ability to absorb mineral elements for the assimilation of organic matter, reflecting the plant’s nutrient use efficiency. In this study, there was no significant difference in the C:N or C:P ratios among different life forms, which contrasts with the results of Liu et al. [17]. This discrepancy may be attributed to the variations in soil nutrient availability and the genetic characteristics of the species studied. Leaf N:P ratio can be used as a diagnostic indicator to determine ecosystems that are subject to nitrogen (N) and phosphorus (P) limitation. Studies have shown that when the ratio of nitrogen to phosphorus (N:P) is less than 14, vegetation growth is limited by nitrogen; when N:P is greater than 16, vegetation growth is limited by phosphorus; and when the N:P ratio falls between 14 and 16, vegetation growth is limited by both nitrogen and phosphorus. In this study, the leaf N:P ratios of various plant communities were all greater than 16 (CG—16.27, GM—17.33, QG—17.81), suggesting that the growth of different plant communities in karst areas is more likely to be limited by phosphorus (P). This aligns with the concept proposed by many scholars that vegetation growth is commonly constrained by phosphorus, especially as vegetation succession progresses [39]. Lower litter C:N ratios are generally considered to indicate high decomposition rates, suggesting faster rates of forest litter decomposition in a region. Among the plant communities with different life types, the grass–shrub community had the lowest C:N ratio (16.68), indicating the fastest decomposition rate of litter and high nutrient cycling efficiency. In contrast, the shrub plant community had the highest C:N ratio (24.25), resulting in the slowest decomposition rate of apomictic litter. The modified results were consistent with the findings of Yu et al. Both studies demonstrated that organic matter decomposition was faster in the grass–shrub and tree–shrub phases. This suggests that in forest communities, maintaining the structural integrity of tree, shrub, and herb hierarchies is essential to enhance microbial utilization of substrates and to increase nutrient turnover and cycling rates [24].

Lower N:P ratios in litter indicate easier decomposition. In this paper, the litter N:P ratio ranged from 19.14 to 33.33, as reported in Pan et al.’s study [40,41]. This suggests that the overall unfavorable decomposition could be attributed to the low nitrogen content and high lignin content in the region and anthropogenic disturbances, as well as the scarcity of soil fauna and microorganisms. These factors contribute to low litter decomposition rates. It can be seen that although karst areas are generally unfavorable to the decomposition of plant communities with different life forms, there are variations among life forms and they are more reliant on soil texture and external disturbances.

Soil ecological stoichiometry is an important indicator for characterizing the composition and quality of soil organic matter [42]. The C:N ratio can be used to assess the rate of decomposition of soil organic matter, while the C:P ratio can characterize the level of effective phosphorus in the soil. Additionally, the N:P ratio is the most effective predictor of nutrient limitation in forests. Compared with global forest soils (14.5, 211, and 14.6) [43] and national terrestrial surface soils (14.4, 136, and 9.3) [44], this study revealed ecological stoichiometries of C:N, C:P, and N:P for forest soils of 14.3, 88.64, and 6.02, respectively. The study found lower C:P and N:P ratios in karst areas with plant communities of various life forms, indicating a high soil P regression level and deficient soil N regression. This suggests that the soil belonged to the N-limited type, further indicating that different plant communities experienced significant habitat pressure [45]. In conclusion, when constructing various living plant communities in karst regions, it is essential to ensure that the layers are complete; enhance the decomposition of apomictic material and the mineralization of soil nutrients; facilitate the accumulation of C, N, P, and other nutrients; and self-regulate their stoichiometric relationships [12]. This will help strengthen their ability to resist fragile habitats.

### 4.3. Ecological Stoichiometric Correlations between Functional Traits of Dominant Species and Leaf Litter–Soil Interactions in Karst Areas with Plant Communities of Different Life Forms

In this study, we found that plant height (PLH), leaf thickness (LT), specific leaf area (SLA), and leaf area (LA) correlated more strongly with soil and litter ecological stoichiometry and were insensitive to leaf ecological stoichiometry. In this study, we demonstrated that the soil C:N ratio was maintained between 10 and 15, which closely resembled the findings of He et al.’s study. This suggests that mineralization and nitrogen release happen rapidly, aiding nutrient uptake by plants. A lower C:N ratio corresponds to a quicker nitrogen release [46]. Combined with our results, the soil carbon-to-nitrogen ratio in karst areas is more sensitive to plant leaf area, leaf thickness, and leaf turnover. In addition, the most favorable soil C:N ratio is usually considered to be around 25:1. When the soil C:N ratio is too high, microbial decomposition and mineralization are slow, and the available nitrogen in the soil needs to be utilized [42]. On the contrary, the C:N ratio is too small in karst areas (CG—14.21, GM—14.53, QG—14.72), resulting in high soil nitrogen contents. To enhance microbial decomposition, it is necessary to increase soil organic matter appropriately to facilitate apoplastic decomposition. The initial C:N ratio of apoplastic matter responds to the impact of climate change on the carbon-to-nitrogen ratio of plants. For instance, nitrogen deposition causes an increase in nitrogen elements, leading to a lower C:N ratio [47]. In our study, apoplastic C:N showed a significant positive correlation with PLH and LA. This suggests that in karst environments, arboreal and shrub plant communities exhibit higher C:N ratios. Plants with high C:N ratios demonstrate increased nitrogen use efficiency, while apoplastic materials with low C:N ratios exhibit fast decomposition traits. Low apoplastic C:N and a wetter climate favor soil organic carbon (SOC) accumulation. This process can be enhanced by both natural processes and human interventions that reduce apoplastic C:N levels. Additionally, global climate change may increase the wetness index, further promoting SOC accumulation [48,49].

Currently, researchers have directed their attention to the impact of soil quality on plant functional traits in forest stands. The findings suggest that soil fertility influences plant functional traits, demonstrating the screening effect of soil elemental ecological stoichiometry on functional traits during the establishment of plant communities [49]. This implies a close relationship between belowground habitats and aboveground components. The reciprocal regulation between the aboveground and belowground components plays a crucial role in shaping ecosystem development and provides the basis for improving vegetation productivity [50]. In this study, a significant positive correlation was observed between soil N:P and SLA, which aligns with the findings of Yu et al. [23]. This may be attributed to the fact that N and P are primary elements composing plant organisms, and the content of N and P also influences the efficiency of photosynthesis, thereby affecting plant growth and physiological state. The findings indicate that the survival and growth of plants are influenced by the stoichiometric balance of limiting elements, and they adapt to various habitats by balancing trade-offs and synergies of functional traits [50,51]. In the future, there will be a comprehensive exploration of the correlation between nutrient element content, microbial population, and biomass, as well as their stoichiometric relationships in the subsurface space, in relation to the functional traits of aboveground plants. This research aims to provide support for the regulation of ecosystem function. 

The findings of this study demonstrated that the association between the functional traits and ecological stoichiometry of plant communities with diverse life forms in karst areas was primarily defined by the interplay between plant functional traits and the ecological stoichiometry of soil and litter. The reasons were analyzed as follows: The high incidence of rocky desertification and habitat heterogeneity in the region, along with the complex karst geological environment, may lead to a greater influence of geological and geographical elements on plant functional traits. These elements include the depth of groundwater, development of fissures, slope, and slope position [8]. However, further exploration is needed to understand the influence of lithology and other geological conditions on plant functional traits. Additionally, factors such as species, succession, and habitat affect the ecological stoichiometry of the leaf–apomictic soil continuum [52]. The genetic characteristics of species play a significant role in determining their selective nutrient absorption and utilization. Additionally, plant communities with varying lifestyles exert regulatory influences on apomictic reserves and stand quality [53]. Consequently, these factors exhibit a strong interdependent relationship with one another. In the future, it is imperative to thoroughly elucidate the mechanism by which habitats influence ecological processes, explore trade-off strategies between different ecological processes, and enhance the self-regulation and self-balancing capacity of ecosystems through moderate disturbance.

## 5. Conclusions

(1)The functional traits of various life forms within plant communities in karst regions demonstrate adaptation to environmental changes through diverse expressions and survival strategies. The growth of various plant communities in karst areas is more likely to be constrained by phosphorus (P), and the potential for apomictic decomposition and soil nutrient mineralization is comparatively weaker.(2)The functional traits of various life forms within karst plant communities exhibit greater sensitivity to soil C: N and apomictic C:N. Disparities in habitat may influence the ecological stoichiometry of the plant leaf–apomictic soil continuum.

## Figures and Tables

**Figure 1 plants-13-02407-f001:**
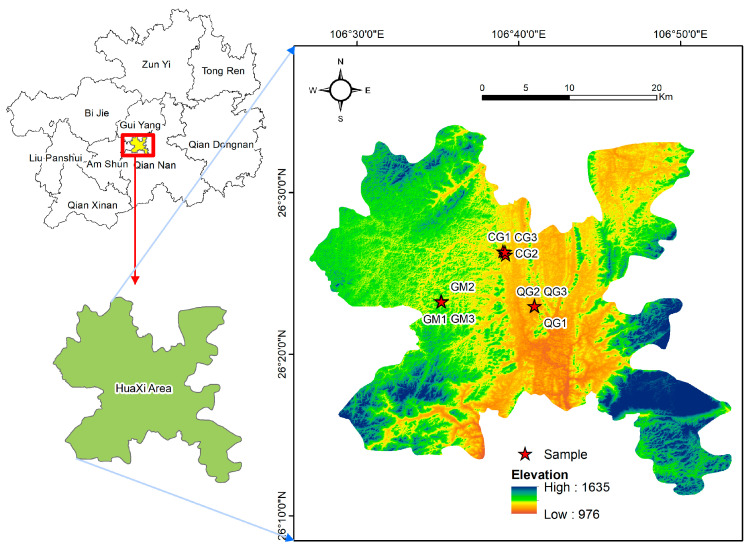
Schematic diagram of the sample area. CG—grass–shrub community, GM—shrub community, QG—tree–shrub community. (The map is original).

**Figure 2 plants-13-02407-f002:**
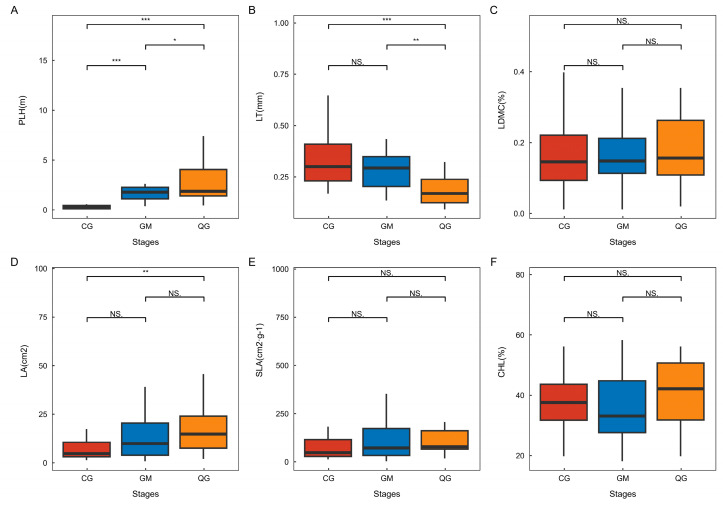
Variations in functional traits of dominant species within plant communities of diverse life forms. In the field of statistics, one-way ANOVA is employed for conducting comparative analysis. * Indicates significant difference (*p* < 0.05), ** indicates highly significant difference (*p* < 0.01) *** indicates highly significant difference (*p* < 0.001). NS signifies no significant difference and lack of ecological statistical significance. Three individuals were measured for each replication, and three replications were conducted for each category of life. Stages are indicative of plant communities characterized by varying life forms. PLH—plant height, LT—leaf thickness, LDMC—leaf dry matter content, LA—leaf area, SLA—specific leaf area, CHL—chlorophyll. The term “CG” denotes a plant community consisting of grass and shrubs, “GM” refers to a plant community dominated by shrubs, and “QG” represents a plant community characterized by a combination of trees and shrubs. The changes in functional traits of plant communities under different life forms are characterized by PLH (**A**), LT (**B**), LDMC (**C**), LA (**D**), SLA (**E**), and CHL (**F**).

**Figure 3 plants-13-02407-f003:**
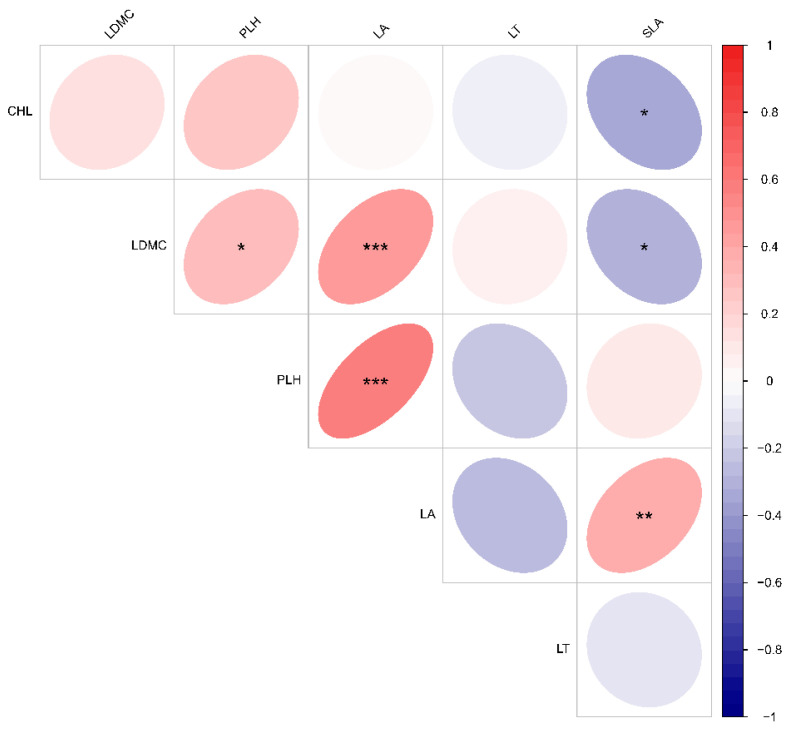
Correlations of functional traits of dominant species in plant communities with varying life forms. PLH—plant height, LT—leaf thickness, LDMC—leaf dry matter content, LA—leaf area, SLA—specific leaf area, CHL—chlorophyll. * Indicates significant difference (*p* < 0.05), ** indicates highly significant difference (*p* < 0.01), *** indicates highly significant difference (*p* < 0.001).

**Figure 4 plants-13-02407-f004:**
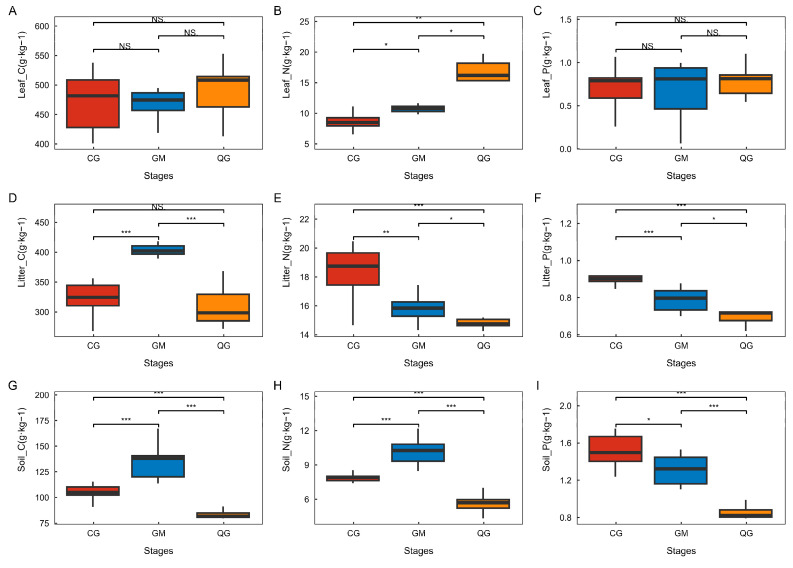
The study examined the carbon, nitrogen, and phosphorus contents of leaf litter and soil for dominant species across various plant-community life forms. In the field of statistics, one-way analysis of variance (ANOVA) is employed for conducting comparative analyses. * Indicates significant difference (*p* < 0.05), ** indicates highly significant difference (*p* < 0.01), *** indicates highly significant difference (*p* < 0.001). Additionally, NS signifies a non-significant difference lacking ecological statistical significance. Each type of life form was subjected to three replications, with three individuals measured in each replication. The stages refer to the plant communities characterized by different life forms, including Leaf_C (leaf carbon content), Leaf_N (leaf nitrogen content), Leaf_P (leaf phosphorus content), Litter_C (litter carbon content), Litter_N (litter nitrogen content), Litter_P (litter phosphorus content), Soil_C (soil carbon content), Soil_N (soil nitrogen content), and Soil_P (soil phosphorus content). These characteristics were observed in CG (grass–shrub plant community), GM (shrub plant community), and QG (tree–shrub plant community). The variables Leaf_C (**A**), Leaf_N (**B**), Leaf_P (**C**), Litter_C (**D**), Litter_N (**E**), Litter_P (**F**), Soil_C (**G**), Soil_N (**H**), and Soil_P (**I**) represent the changes in carbon, nitrogen, and phosphorus contents of leaves, dead litter, and soil in plant communities with different life forms.

**Figure 5 plants-13-02407-f005:**
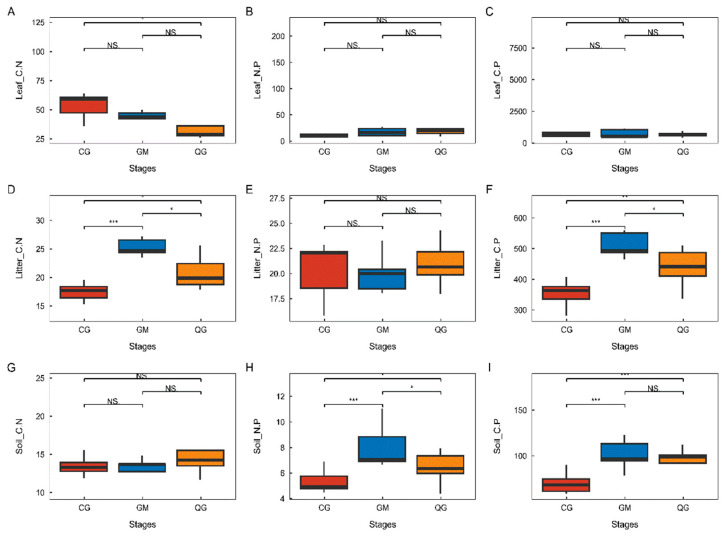
Alterations in the ecological stoichiometry of leaf litter and soil for predominant species within plant communities exhibiting diverse life forms. In statistical analysis, one-way ANOVA is employed to compare groups. * Indicates significant difference (*p* < 0.05), ** indicates highly significant difference (*p* < 0.01), *** indicates highly significant difference (*p* < 0.001), NS signifies a non-significant difference lacking ecological statistical significance. Each life form was subjected to three replicates, with measurements taken for three individuals in each replicate. The study examined the variations in the leaf–litter–soil ecological stoichiometry of plant communities with different life forms, including stages of plant communities and their associated leaf, litter, and soil characteristics. These characteristics include Leaf_C.N (**A**), Leaf_N.P (**B**), Leaf_C.P (**C**), Litter_C.N (**D**), Litter_N.P (**E**), Litter_C.P (**F**), Soil_C.N (**G**), Soil_N.P (**H**), and Soil_C.P (**I**). The plant communities were categorized into CG (grass–shrub), GM (shrub), and QG (tree–shrub) plant communities.

**Figure 6 plants-13-02407-f006:**
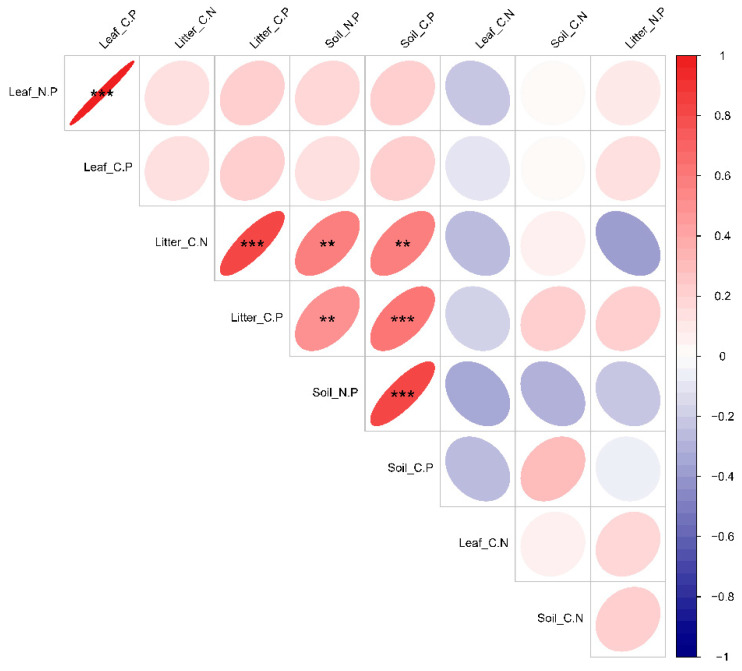
The study examined the correlation between the ecological stoichiometry of leaf, litter, and soil for dominant species within plant communities of various life forms. The variables Leaf_C.N, Leaf_N.P, and Leaf_C.P denote leaf carbon–nitrogen content, leaf nitrogen–phosphorus content, and leaf phosphorus content, respectively. Similarly, Litter_C.N, Litter_N.P, and Litter_C.P denote the contents of carbon–nitrogen, nitrogen–phosphorus, and phosphorus in litter, respectively. Soil_C.N, Soil_N.P, and Soil_C.P denote the levels of carbon–nitrogen, nitrogen–phosphorus, and phosphorus in the soil, respectively. The plant communities under investigation comprised CG (grass–shrub), GM (shrub), and QG (arboreal–shrub) communities. ** and triple *** symbols indicate highly significant differences at *p* < 0.01 and *p* < 0.001, respectively.

**Figure 7 plants-13-02407-f007:**
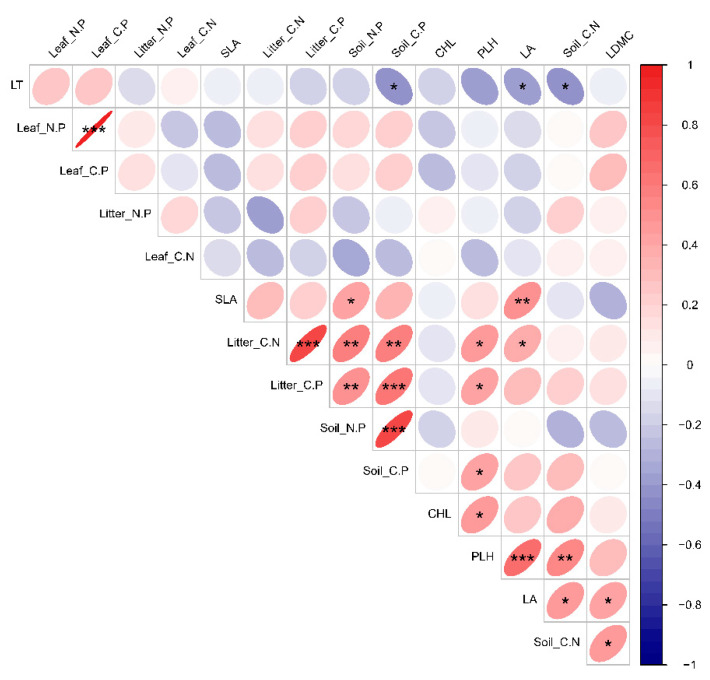
The study examined the relationship between the functional traits of dominant species in plant communities and the ecological stoichiometry of leaf litter and soil. The functional traits comprised plant height (PLH), leaf thickness (LT), leaf dry matter content (LDMC), leaf area (LA), specific leaf area (SLA), chlorophyll content (CHL), leaf carbon content (Leaf_C.N), leaf nitrogen content (Leaf_N.P), leaf phosphorus content (Leaf_C.P), litter carbon content (Litter_C.N), litter nitrogen content (Litter_N.P), litter phosphorus content (Litter_C.P), soil carbon content (Soil_C.N), soil nitrogen content (Soil_N.P), and soil phosphorus content (Soil_C.P). The plant communities under investigation included CG (grass–shrub), GM (shrub), and QG (tree–shrub) communities. Refer to comments on Figure 3. * Indicates a statistically significant difference at *p* < 0.05; ** and triple *** symbols indicate highly significant differences at *p* < 0.01 and *p* < 0.001, respectively.

**Figure 8 plants-13-02407-f008:**
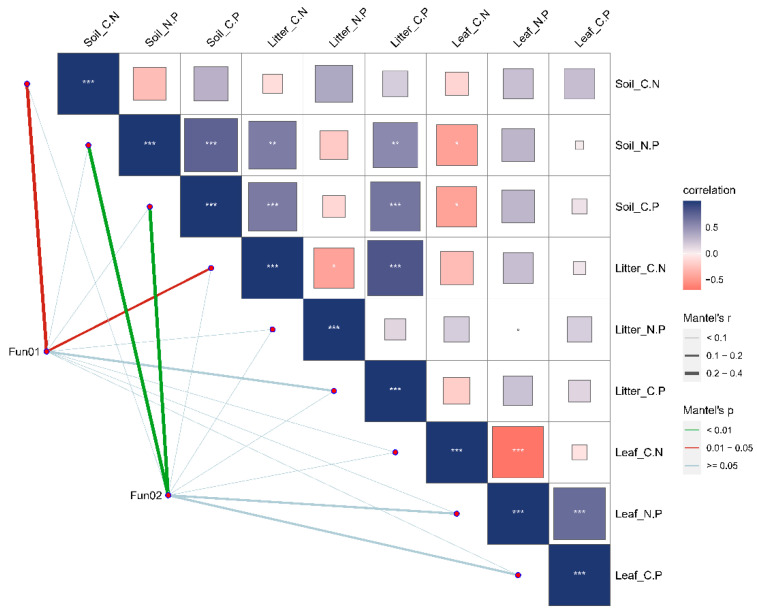
Mantel analysis of the functional shapes of dominant species in plant communities and deciduous leaf–soil ecological stoichiometry. The figure illustrates pairwise comparisons of functional shapes with ecological stoichiometry for various plant life forms, with color gradients representing Spearman’s correlation coefficients. The width of the edge corresponds to the Mantel r statistic for distance correlation, while the color of the edge indicates the statistic based on 9999 alignments. * Indicates a statistically significant difference at *p* < 0.05; ** and triple *** symbols indicate highly significant differences at *p* < 0.01 and *p* < 0.001, respectively.

**Figure 9 plants-13-02407-f009:**
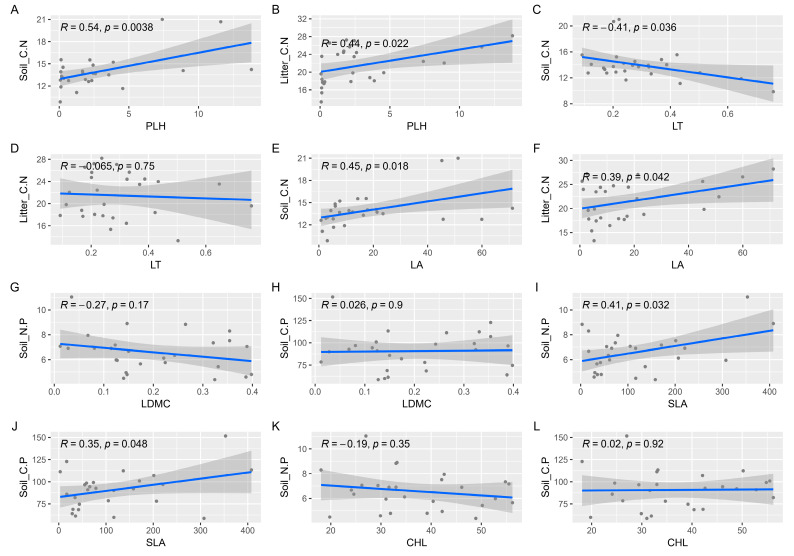
A line plot illustrating the relationship between ecological stoichiometry and functional traits. R represents the magnitude of correlation between the two variables, while *p* represents the level of significance. When *p* is less than or equal to 0.05, it signifies a statistically significant linear correlation between the two variables. The study investigated the relationships between various plant traits and soil properties. The plant traits included plant height (PLH), leaf thickness (LT), leaf dry mass (LDMC), leaf area (LA), specific leaf area (SLA), and chlorophyll content (CHL). The soil properties examined were soil C:N, litter C:N, soil N:P, and soil C:P. The study found linear correlations between PLH and soil C:N (**A**), litter C:N (**B**), LT and soil C:N (**C**), litter C:N (**D**), LA and soil C:N (**E**), litter C:N (**F**), LDMC with soil N:P (**G**), soil C:P (**H**), SLA with soil N:P (**I**), soil C:P (**J**), CHL with soil N:P (**K**), and soil C:P (**L**). The dots represent sample data, blue lines represent correlation trends, and gray areas represent confidence intervals.

## Data Availability

The original contributions presented in the study are included in the article, further inquiries can be directed to the corresponding author.

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
