# Peer review of "Influence of Functional Traits of Dominant Species of Different Life Forms and Plant Communities on Ecological Stoichiometric Traits in Karst Landscapes"

_plants, 2024, doi:10.3390/plants13172407_

Round 1

Reviewer 1 Report (Previous Reviewer 5)

Comments and Suggestions for Authors

Very important  information is missing in the results: ranges of absolute values of C, N, and P in the plant leaves, soil, and litter, also ranges of ratios of the absolute values of the mentioned rations should be provided. Without absolute values all data about correlations between plant leaf – litter – soil are not based experimentally. The most valuable ratios are not provided: c ratio between leaf and litter, or C ratio between leaf and soil, the same concerns N ratios between plant leaf and litter, or plant leaf and soil, or P ratios, and only after declaration of them, there is sence to analyse rations of different elements and subsequently correlations.  The design of the selected studies is very complex and the authors themselves do not seem to be fully aware of which „ecological stoichiometric indicators“ (speaking simply and clearly, which element ratios) are the most important distinquishing types of vegetation.

Comments on the Quality of English Language

I am not an expert

Terminology is problem from biology point of view

Author Response

Response to Reviewer 1 Comments

Thank you very much for taking the time out of your busy schedule to review this manuscript. We appreciate all your comments and suggestions. We have revised the manuscript and response your questions in highlight, please see the revised manuscript. Here were the main questions.

Q1: Very important  information is missing in the results: ranges of absolute values of C, N, and P in the plant leaves, soil, and litter, also ranges of ratios of the absolute values of the mentioned rations should be provided. Without absolute values all data about correlations between plant leaf – litter – soil are not based experimentally. The most valuable ratios are not provided: c ratio between leaf and litter, or C ratio between leaf and soil, the same concerns N ratios between plant leaf and litter, or plant leaf and soil, or P ratiosand only after declaration of them, there is sence to analyse rations of different elements and subsequently correlations.  The design of the selected studies is very complex and the authors themselves do not seem to be fully aware of which „ecological stoichiometric indicators“ (speaking simply and clearly, which element ratios) are the most important distinquishing types of vegetation.

Response: We are very grateful to the reviewers for raising such doubts. Firstly, the absolute ranges of C, N, and P in plant leaves, soil, and litter were not presented in the manuscript, but we visualized them graphically. We acknowledge the reviewer's suggestion that the absolute ranges of C, N, and P in plant leaves, soil, and apoplast should be detailed in the results section. Accordingly, we have revised this information in the results analysis section of the manuscript. Secondly, our experiments were designed to reveal the ecological stoichiometries among plant leaves, soil, and apoplasts, which may have appeared less concise in some descriptions, for which we sincerely apologize. Lastly, to differentiate vegetation types based on "ecological stoichiometries, we focused on C:N, C:P, and N:P ratios of plant leaves, soil, and litter as crucial ecological stoichiometries. We have also refined this section in the manuscript to enhance readability and objectivity.

In addition, we have made revisions to the manuscript based on the reviewers' comments, and the updated content is highlighted in the revised manuscript. (L330, L332-333, L334-336, L341-343, L346-348, L484, L489-492, L533-534)

We would like to express our gratitude to the reviewers for their valuable suggestions on the manuscript, which we find to be highly constructive!

Finally, we are very grateful to the reviewer for reading the manuscript and putting forward very meaningful suggestions. This is a great help to our article, and we are deeply grateful for it.

Reviewer 2 Report (Previous Reviewer 4)

Comments and Suggestions for Authors

Dear Authors,

Your contribution is interesting and well-prepared article on the effectiveness of two modern approaches (plant functional traits and soil-ecological stoichiometry) for determination of plant-soil interactions. The results demonstrate existence of modest correlation between attributes of both approaches. Perhaps it is because functional traits in the article are a little bit far of parameters reflecting plant productivity and composition of plant communities. However, my conclusion: the article shall be published after correction of listed below drawbacks.

Comments

There is a tautology in the title (twice “communities”), please edit.

L. 34  “apomictic decomposition”. Term “apomictic” is not in use in forest ecology and soil science (in the meaning as in this article). The authors did not give its definition in the text. I am investigating soil decomposition processes during decades, but I met “apomictic” first time. Direct meaning of “apomictic decomposition” is a full oxymoron: “non sexually reproducing decomposition”. I guess the authors mean that it is vegetative organs of plants: leaves, twigs, branches.   Isn’t it?    Please give a comprehensive definition of this new term or remove it from the whole text because terminological manipulations are clear indicators of “pseudo novelty” in science.

L 111-113. According to the content of the article, article’s aim (…to uncover the functional traits and ecological stoichiometric characteristics of various life-types within plant communities…) means “a study of the impact of vegetation on soil”, or opposite: “impact of soils on vegetation”. It is my personal opinion.

L.112. “life-types” - is it “morphological life form” in ecological botany? In second part of the text, authors use “life form”. Please correct the whole text.

L. 118-128.  2.1. Study area. It is unsatisfactory for international journal. Please obligatory add here, in Study area (by moving from other parts of the text), (a) description of karst vegetation (names of plant species, plant communities or forest types with few words on their density, structure and biomass); (b) soils’ mosaics as a proportion of different soils in the karst landscape, soil taxonomic names according WRB or American classifications, dominating soil profile types, consequence and thickness of soil horizons and whole profile down to limestone, basic physical and chemical parameters.

L. 132 – 134 and 178-185 shall be in the Study area.

L. 191.  “depth” –is it  soil depth?

L 194.  “litter”  please specify: is it a fresh litter fall or soil organic layer as grass sward or forest litter, forest floor?

L 200-203. Please add dimensions for all parameters, for example (PLH, m), (LT, mm) etc.

L.258. Fig 2 and other figures. Why scale of plant communities has name “Stages”? There is no discussion on dynamic aspects of vegetation development in Section 2.

L 266. Fig 2. Dimension of Life area, LA, is cm2:  is it per plant, per m2?  Then:  LDMC, %  - per cent of what? Please specify.

Sincerely yours

Reviewer

Author Response

Response to Reviewer 2 Comments

Thank you very much for taking the time out of your busy schedule to review this manuscript. We appreciate all your comments and suggestions. We have revised the manuscript and response your questions in highlight, please see the revised manuscript. Here were the main questions.

Q1:There is a tautology in the title (twice “communities”), please edit.

Response: We accepted the reviewers 'comments and have revised the manuscript.(L1-4)

Q2:L. 34  “apomictic decomposition”. Term “apomictic” is not in use in forest ecology and soil science (in the meaning as in this article). The authors did not give its definition in the text. I am investigating soil decomposition processes during decades, but I met “apomictic” first time. Direct meaning of “apomictic decomposition” is a full oxymoron: “non sexually reproducing decomposition”. I guess the authors mean that it is vegetative organs of plants: leaves, twigs, branches.   Isn’t it?    Please give a comprehensive definition of this new term or remove it from the whole text because terminological manipulations are clear indicators of “pseudo novelty” in science.

Response:Thank you very much to the reviewers for providing such professional opinions. We did make mistakes in our language expression. Based on the reviewer's opinion, we believe that replacing "litter decomposition" can more accurately express the content of the article.(L34)

Q3:L 111-113. According to the content of the article, article’s aim (…to uncover the functional traits and ecological stoichiometric characteristics of various life-types within plant communities…) means “a study of the impact of vegetation on soil”, or opposite: “impact of soils on vegetation”. It is my personal opinion.

Response: We are very impressed with the reviewers' point of view. By studying the characteristics of plant functionality and ecological stoichiometry, we have finally clarified the interaction between soil and vegetation. We have incorporated these revisions into the manuscript.(L113)

Q4:L.112. “life-types” - is it “morphological life form” in ecological botany? In second part of the text, authors use “life form”. Please correct the whole text.

Response: What we want to present are different life forms of plants. This may be an error in our expression. Based on the reviewer's opinion, we have revised it in the manuscript. (L112)

Q5:L. 118-128.  2.1. Study area. It is unsatisfactory for international journal. Please obligatory add here, in Study area (by moving from other parts of the text), (a) description of karst vegetation (names of plant species, plant communities or forest types with few words on their density, structure and biomass); (b) soils’ mosaics as a proportion of different soils in the karst landscape, soil taxonomic names according WRB or American classifications, dominating soil profile types, consequence and thickness of soil horizons and whole profile down to limestone, basic physical and chemical parameters.

Response: In fact, we had already taken this issue into consideration before finalizing the manuscript. After carefully reviewing the feedback from the reviewers, we deemed it essential to incorporate revisions based on their suggestions. Consequently, we meticulously revised this section of the manuscript.(L131-156)

Q6:L. 132 – 134 and 178-185 shall be in the Study area.

Response: We couldn't agree more. We have already completed the revision of the manuscript. (L131-156)

Q7:L. 191.  “depth” –is it  soil depth?

Response: Yes, it represents soil depth.

Q8:L 194.  “litter”  please specify: is it a fresh litter fall or soil organic layer as grass sward or forest litter, forest floor?

Response: The litter in this study refers to the naturally withered plant leaves due to seasonal changes.

Q9:L 200-203. Please add dimensions for all parameters, for example (PLH, m), (LT, mm) etc.

Response: We have improved Figure 2.(L272)

Q10:L.258. Fig 2 and other figures. Why scale of plant communities has name “Stages”? There is no discussion on dynamic aspects of vegetation development in Section 2.

Response: We are pleased to address the reviewer's question. The "stages" mentioned in the manuscript actually correspond to the life forms of the plant community. Various stages of a plant community exhibit distinct life forms, hence we utilize "stages" to depict them in Figure 2. In terms of the description of this section of the results, our emphasis is on comparing the variations in functional traits of plant communities across different life forms, rather than conducting a dynamic evaluation.

Q11:L 266. Fig 2. Dimension of Life area, LA, is cm2:  is it per plant, per m2?  Then:  LDMC,  - per cent of what? Please specify.

Response: First, LA represents leaf area, which indicates the leaf area of each plant. Secondly, LDMC represents the dry matter content of leaves, and its calculation formula is: (leaf dry weight - leaf wet weight) / dry weight * 100%. Therefore, the unit of LDMC is %.

Finally, we are very grateful to the reviewers for their numerous constructive comments on the manuscript, which have enhanced the quality and readability of the document. Thank you once again!

Round 2

Reviewer 1 Report (Previous Reviewer 5)

Comments and Suggestions for Authors

Excellent changes have been done

Just there is need to finalise unification: everywhere using "litter"

 leaf-litter-soil

leaf-debris-soil

leaf-decay-soil

leaf decay

row 227: Plant leaf area (PLH) is it wright?

Author Response

Response to Reviewer 1 Comments

Thank you very much for taking the time out of your busy schedule to review this manuscript. We appreciate all your comments and suggestions. We have revised the manuscript and response your questions in highlight, please see the revised manuscript. Here were the main questions.

Q1:Just there is need to finalise unification: everywhere using "litter"

 leaf-litter-soil

leaf-debris-soil

leaf-decay-soil

leaf decay

Response: We uniformly use "litter" in manuscripts and We have revised it in the manuscript.

Q2:row 227: Plant leaf area (PLH) is it wright?

Response: We are very sorry for this error. We have revised it in the manuscript. This should be LA. (L227)

Finally, we are very grateful to the reviewer for reading the manuscript and putting forward very meaningful suggestions. This is a great help to our article, and we are deeply grateful for it.

Reviewer 2 Report (Previous Reviewer 4)

Comments and Suggestions for Authors

Dear authors,

Your article is now ready for publication.

Sincerely yous

Reviewer

Author Response

Response to Reviewer 2 Comments

We are very grateful to the reviewers for proposing revisions to this manuscript, and thank you very much for your approval of this article!

This manuscript is a resubmission of an earlier submission. The following is a list of the peer review reports and author responses from that submission.

Round 1

Reviewer 1 Report

Comments and Suggestions for Authors

Please find the attached review report (MS Word document)

Comments on the Quality of English Language

Only some minor issues in choice of word needs to be taken care of

Reviewer 2 Report

Comments and Suggestions for Authors

The fact that functional plant traits are related to environmental factors and the structure of plant communities is well known. What is really new in this study? It lacks a clear hypothesis that has been tested. The results are primarily descriptive. It is also completely unclear whether the spectrum of the respective species in the three plant communities was tested or whether only dominant species were compared. It is well known that herbaceous species, shrubs and trees differ in functional characteristics! We always speak of life forms. You don't even get an overview of the life form spectra of the plant communities compared. Were individual species compared with each other or the plant communities?  Many ambiguities need to be clarified!

Reviewer 3 Report

Comments and Suggestions for Authors

The manuscript with the title “Influence of functional traits of dominant species of different life forms and plant communities in karst on leaf-litter-soil ecological stoichiometric traits” presents the relationship between the functional characteristics of dominant species in plant communities of various life forms from karst environments. The study focused on leaf-deciduous-soil ecological stoichiometry. Authors found considerable variability in the functional trait characteristics of dominant species across different plant community life forms.

The abstract is a little bit long. I suggest to keep only most relevant and novel/interesting things in the abstract.

Figure 1. If the map is original shall be said in the caption (original), if it was taken from a source authors have to give the source.

Lines 117-193. For some species the author is given (e.g. Quercus fabri Hance), while for some not (e.g. Myrsine africana). Authors should decide to either give the author to all species (using a reputable and update source e.g. IPNI) or do not give the author after none of the species. Some name are out of date: e.g. Conyza canadensis is now Erigeron canadensis. I suggest a taxonomic update as well, using a good source (e.g. https://powo.science.kew.org/results?).

Conclusions are structured 1), 2) and 3). These should correspond to 3 objectives that shall be given at the end of the introduction.

Best regards.

Comments on the Quality of English Language

minor English style improvements might be needed.

Reviewer 4 Report

Comments and Suggestions for Authors

Dear authors,

Your article looks like a comprehensive study of plant-soil relationships using a concrete example of vegetation and soils in karst landscape. However, it leaves a strange impression and many open questions. The structure of the article is a classic scheme “experimental data - correlations – regressions”. It was widely used in ecological botany, forest ecology, forest inventory and now in ecological monitoring but here it is under banner of functional traits and ecological stoichiometry. A strange thing is an absence of citations on works followed to above-mentioned scientific fields.

Important. “2.1  Overview of the study area” (L 116-126):  10 lines only!  It is nothing! There are here no any scientific names and standard description of plant associations (submitted to journal PLANTS!), no scientific soil taxonomic names and no soil physical and chemical parameters of the sample plots under investigation. Dominated plant species are mentioned only on lines 177-183 but they shall be here in sites’ description. The edaphic site (biom, habitat, landscape unit) is characterized by one sentence (L 125):“The study area is characterized by a limestone soil matrix, with the predominant soil types being yellow limestone soil and black lime soil [24]”. By the way,   ….”soil matrix” …. I think “soil spatial mosaics” is a more precise definition.  The article looks as a story on separate parameters’ interactions for any unknown natural ecosystems.

Additional comment to Section 2:   Elements‘ concentrations are not enough for the study on comparative contribution of various elements in plant growth, productivity and soil development. They shall be now mass pools and flows of elements related to area and time. Please discuss this circumstance.

Figures 2, 4, and 5 with experimental data are the more valuable part of this contribution.

Figures 3, 6, 7, and 8  with “pale” correlations are not expressive.

Figure 9 demonstrates very poor linkage between functional traits of plants and ecological stoichiometry of soils, actually between plant and soil. But it is not true. I think it shows a low effectiveness of authors’ theoretical and methodological approach.

The Discussion has been written without comparison with previous researches and without accentuation the novelties in this work.

Functional traits are not discussed and specified for all or selected dominant species.

NB! All Figures with graphs  and legends shall be corrected:  on the ordinate (vertical) axises, please obligatory(!) add dimensions of values for all parameters (for example meter, %,  mg kg-1 and so on).

The text has too many introducing sentences on general scientific postulates (just as from textbooks), it is very verbose and should be significantly shortened.

The declared in the title task (“Influence of functional traits…”) actually is not solved and the influence stays disputable.  

Specific comments

Title:       “karst landscape” is better

L 73  “ecological chemometrics”  is not defined in the text.

L 113  “The objective is to furnish references for research on the mechanisms involved  in the formation of forest communities and the preservation of biodiversity”.  The References are too short and content of this article is not so expressive to be of interest for researchers. I advise deleting this sentence.

L 130      “Three distinct plant communities in the region, namely the grass-shrub plant com-munity (CG), the shrub plant community (GM), and the tree-shrub plant community (QG), were selected for the study based on the findings of previous research” It is blend definitions without scientific background (see comment above)

L 174  “relative abundance, relative frequency, and relative cover”,    what is a scale? 5, 10, 100?

L 186  Please correct brackets for citation (26)

L 198      ”...functional diversity of production…”  Functional diversity of production should be measured ONLY by the rates of processes (flows) but never by the values of state variables  =  functional traits

L 199 “plant height (PLH)” or “plant leaf height (PLH)” (L 438)?  If second, please give more detail what it means and how to measure it.

L 199  „…leaf thickness (LT), …. leaf dry matter content (LDMC), leaf area (LA), and specific leaf area (SLA)”.  Is it for only one representative leaf?         Leaves of trees, grasses and forbs of various plant species are strongly different by form, weight and other parameters. Why you give no word how you take all this into account?

L 224   2.5. Data processing:  please add sources of data for all used software. Mantle analysis even not mentioned here.

L 415  “….the relationship between ecological stoichiometry and functional shape. "   shape  or   traits?

L 436. “These findings indicate that plants of varying life forms demonstrate unique leaf characteristics in order to adapt to the same environment [30].”  -  It is news from plant morphology of 19 century.

L 438. “The study found that the plant leaf height (PLH) of the arborvitae community was notably greater than that of the shrub and grass shrub communities (Figure 2-A).  This suggests that the arborvitae community exhibits greater competitiveness, productivity, and recovery ability compared to the shrub and grass-shrub communities”. It is only authors’ opinion without experimental background and without citation of previous researches.

L 468-472 “The leaf N:P ratios of plant communities representing various life forms in this study were all observed to exceed 16, indicating that the growth of plant communities in karst areas is  more likely to be constrained by phosphorus (P). This discovery (but it is only well known fact!) is consistent with the concept advanced by the majority of scholars, which suggests that vegetation growth is readily limited by phosphorus during the succession of vegetation [34]”.          It is true only for grasslands, subtropical and tropical forests. In boreal and temperate climate,  N limitation is a dominating factor. Please specify.

Reviewer 5 Report

Comments and Suggestions for Authors

Title

In the title and the text below title „leaf-litter-soil ecological stoichiometric traits“ is not logistic:  leaf-litter-soil stoichiometric traits“ more than enough. The other possible phrase „ecological stoichiometric traits of communities“ The words leaf and soil themselves indicate that they refers to the subject of the science of plant ecology - an organism's relationship to its environment

 Mystified title somehow should be changed to informative title like “Comparison of C, N, P ratios in leaf-litter-soil between communities with different life forms (case  study in Guizhou, China)”

Visibility of such experimental analysis is very low according to the provided style of the title and all the text. “Stoichiometry and functional traits” would be relevant title of the world wide study.

 Abstract

Statement  „This approach offers a more comprehensive understanding of ecosystem processes and functions in contrast to the examination of taxonomic diversity of species.“ contradicts to the research subject – it is investigation of species (not all, only dominant). „The findings of our study revealed considerable variability in the functional trait characteristics of dominant species across different plant community life forms.“ = other formulation is required.

  in Guizhou – what is it – town, city, region of China? - most international readers are not familiar with this title. In the abstract the location of the area in China should be provided, also area size,  and the most important dominant species of the communities, analysed life forms

 „positive correlations between soil N: P, withered material C: N, and apomictic C: P“ – absolutely unclear, some atypic terminology „withered“, „apomictic“, „stages“ in the figures

  Furthermore, our study found that plant height (PLH), leaf area (LA), and specific leaf area (SLA) were particularly sensitive to the ecological stoichiometry of soil and apomictic.

In the abstract so many different „apomictic“: apomictic, apomictic decomposition, apomictic material. It is confusing – everyone knows „apomixis“, although what does it means „apomictic“ – it is absolutely unclear.

The abstract does not provide concentrated in one sentence information about exact estimations which have been done – what elements have been examined in leaves-litter-soil,

 for example - soil N: P – well known topic since long time, according to the title the article stoichiometry would be ratio comparing soil N:P - litter N:P - leaf N:P, etc. in various selected species of different life forms.

Information is missing about functional traits of diverse plant community life forms which were investigated in present study     

 „Specifically, strong positive correlations were observed among the plant height (PLH), leaf area (LA), leaf dry matter content (LDMC), and specific leaf area (SLA) of the dominant species. – old, well described discoveries in plant growth literature about relationships of morphophysiological traits, what is new?

Such abstract – philosophy, which might be written without any research, it is mist. The abstract should be rewritten totally, in brief form providing essential results, based on statistics. The authors of the article are unable to summarise how they did their job and what they gained.

 Introduction

 Exact definition of “functional traits” should be provided, what is assumed as functional trait, what difference is between morphological and functional traits? – is plant height – functional trait? – firstly the authors should provide exact reference for such definition. Any origin feature of the organism is titled under ‘functional trait”? At the beginning of paragraph functional traits - „ functional traits, which includes the examination of leaf, branch, and stem traits“, at the end of paragraph – „Various indicators of plant functional traits, such as specific leaf area, leaf nitrogen content, leaf thickness, leaf dry matter content, chlorophyll content, leaf relative water content, stem tissue density, leaf carbon content, stem nitrogen content, stem phosphorus content“.

 International content is missing in the introduction – “Plants” – is devoted for international reader. According description it looks that no one else has done stoichiometric investigations.

The citing is very formal; it does not provide state-of-the art before investigation described in the paper.

The objective should have exact statements  („is to furnish references for research on the mechanisms involved in the formation of forest communities and the preservation of biodiversity“ –„Plants“ is a journal for rigorous, named biological research, not for philosophy that is not based on research results.

 Methods

„research methodology“- it is not accepted practice of the international journals

Description of plot size (2.2 paragraph) should be illustrated by compound figure (3 location map is insufficient). Coordinates and related climatology data should be provided for each community separately.

Dominant species lists for each community should be provided.

 Results

Numeric data are missing – bases for correlations are unknown. Without such data content looks as falsified and mystified. Without numerical data all correlations do not have any value. Such behavior source for plagiarism of the results in the next publications.  

 Three tables with numeric data (mean plus CD or CI) should be provided in the results:

Table 1. Morphological and physiological traits of 27 species listed.

Table 2. Proportion of the dominant species inside each community (for all three communities).

Table 3. Ratios of C-P, C-N, N-P in leaf-litter-soil of all three communities

 Figure titles are too belletrist, at the same time required information is missing.

Stages – what does it mean? It should be removed and full one-two word titles of the communities in the graphs should be provided, also besides compound figure a), b) etc., titles should be provided. Reading of the article in present form – a waste of time for the reader without any benefit.   

 Figures are not described, it means results do not have descriptions – The authors leave the analysis of the results to the reader, which is none of their business.

 Discussions

Interpretation of each figure is missing. International research based discussions are missing.

Comments on the Quality of English Language

Relevant international terminology should be used, belletristic removed from the titles of the figures